# Twelve-Year Single Center Experience Shows Safe Implementation of Developed Peritoneal Surface Malignancy Treatment Protocols for Gastrointestinal and Gynecological Primary Tumors

**DOI:** 10.3390/cancers13102471

**Published:** 2021-05-19

**Authors:** Philipp Horvath, Can Yurttas, Stefan Beckert, Alfred Königsrainer, Ingmar Königsrainer

**Affiliations:** 1Comprehensive Cancer Center, Department of General, Visceral and Transplant Surgery, University of Tübingen, 72076 Tübingen, Germany; philipp.horvath@med.uni-tuebingen.de (P.H.); can.yurttas@med.uni-tuebingen.de (C.Y.); Stefan.Beckert@sbk-vs.de (S.B.); alfred.koenigsrainer@med.uni-tuebingen.de (A.K.); 2Department of General and Visceral Surgery, Schwarzwald-Baar Klinikum, 78052 Villingen-Schwenningen, Germany; 3Department of General, Visceral and Thoracic Surgery, Academic Teaching Hospital Feldkirch, Carinagasse 47, 6800 Feldkirch, Austria

**Keywords:** peritoneal metastases, morbidity, outcome, HIPEC, cytoreductive surgery

## Abstract

**Simple Summary:**

The treatment of peritoneal surface malignancies (PSM) has dramatically evolved during the past two decades. Indications, treatment protocols, surgical techniques and the application of HIPEC in the prophylactic setting were evaluated in the surgical community. Nevertheless, the current results of the PRODIGE-7 trial disfavored the application of HIPEC for PSM of colorectal cancer and raised uncertainty among surgeons. On the other hand, cytoreductive surgery and HIPEC represent state-of-the-art therapy for peritoneal mesothelioma (except the sarcomatoid-subtype) and pseudomyxoma peritonei. Comparing the literature is cumbersome due to the variety of HIPEC protocols and differences in indication settings. This article aims to provide an insight into the impact of different HIPEC protocols, different indication settings and the implementation of pre-HIPEC laparoscopy on patients’ morbidity rates and outcomes and serves as guidance for surgeons dealing with these patients in order to guarantee high-quality treatment.

**Abstract:**

(1) *Background*: Cytoreductive surgery and hyperthermic intraperitoneal chemotherapy provide survival benefits to selected patients. We aimed to report our experience and the evolution of our peritoneal surface malignancy program. (2) *Methods*: From June 2005 to June 2017, 399 patients who underwent cytoreductive surgery plus hyperthermic intraperitoneal chemotherapy at the Tübingen University Hospital were analyzed from a prospectively collected database. (3) *Results*: Peritoneal metastasis from colorectal cancer was the leading indication (group 1: 28%; group 2: 32%). The median PCI was 15.5 (range, 1–39) in group 1 and 11 (range, 1–39) in group 2 (*p* = 0.002). Regarding the completeness of cytoreduction (CC), a score of 0 was achieved in 63% vs. 69% for group 1 and 2, respectively (*p* = 0.010). Median overall survival rates for patients in group 1 and 2 for colon cancer, ovarian cancer, gastric cancer and appendix cancer were 34 and 25 months; 45 months and not reached; 30 and 16 months; 39 months and not reached, respectively. The occurrence of grade-III and -IV complications slightly differed between groups (14.5% vs. 15.6%). No 30-day mortality occurred. (4) *Conclusions*: Specialized centers are able to provide low-morbidity cytoreductive surgery and hyperthermic intraperitoneal chemotherapy without mortality. Strict patient selection during the time period significantly improved CC scores.

## 1. Introduction

The management of peritoneal surface malignancies (PSM) has experienced a crucial change in recent years. Once an inoperable and palliative situation, with the only therapeutic option being systemic chemotherapy, there is now a curative treatment that can be provided for a selected patient subset. The acclaim of hyperthermic intraperitoneal chemotherapy (HIPEC) subsided after results of the PRODIGE-7-, PRODIGE 15- and COLOPEC-trial [1,2,3] were presented. The high-dose and short-term oxaplatin (OX)-based HIPEC is now widely abandoned, and other HIPEC compounds, such as cisplatin and mitomycin C (MMC), have regained attention. Above all, the PRODIGE-7 trial was criticized due to a variety of methodological weaknesses, including an expected overoptimistic survival in the experimental arm, combined with an underestimation of the effect of cytoreductive surgery alone [4]. Furthermore, the short drug exposure time of OX, the possible adverse effects of the carrier solution (Dextrose 5%) and possible adverse effects of hyperthermia itself were considered, explaining the negative results of the study. The same is true for the PRODIGE-15 study, which was mainly criticized due to a highly heterogenous study population (non-metastatic and metastatic setting) [5].

To date, no randomized controlled trial comparing the efficacy of different HIPEC compounds exists, but a recent meta-analysis, including 11 studies and 2091 patients, comparing MMC- and OX-based HIPEC, suggested no evidence for differences in overall and disease-free survival but a statistically significantly increased rate of major complications to the disadvantage of OX [6].

Undoubtedly, the value of a high-quality cytoreductive surgery (CRS) is still a crucial cornerstone in order to provide satisfactory survival rates for patients with PSM. The PRODIGE-7 trial, despite the lack of efficacy of additional HIPEC, showed a median overall survival rate of 41 months, bearing in mind that approximately 25% of patients had PCI scores (peritoneal cancer index) >16, and patients with signet-ring histology were also included. During the recruitment periods of the abovementioned randomized trials, indications in favor of CRS and HIPEC, for almost all tumor etiologies, narrowed. On the other hand, evidence grew, that concomitant colorectal liver and peritoneal metastases should not contraindicate combined surgery and HIPEC for a strictly selected patient subset, here defined by their response to systemic chemotherapy, a low to moderate PCI (<17) and a maximum of three good resectable liver metastases (LM) [7]. Apart from the PCI score, a variety of tumor-, patient- and treatment-specific parameters now impact the treatment algorithm. A favorable PCI (colorectal: <16 [8]; gastric: <6–10 [9]; ovarian: not defined yet [10,11,12]), a favorable histology (signet-ring histology is considered a relative contraindication for CRS and HIPEC [13,14]) and response to systemic chemotherapy should be present. These parameters should allow for a proper patient selection and, in combination with a structured perioperative complication management, acceptable morbidity and mortality rates should be achievable. Recent data have shown that, over time, a lower percentage of patients died due to manageable but potentially life-threatening complications after CRS and HIPEC, suggesting a higher expertise in patient selection and complication management [15]. Furthermore, data showed that cumulative in-hospital mortality after CRS and HIPEC over a 9-year period was only 3.4% in Germany, which is lower than for oncologic pancreas, esophagus and liver surgery [15]. Center experience is another factor impacting on postoperative morbidity [16] after CRS and HIPEC, emphasizing the importance of high-volume centers.

This whole evolution process of PSM treatment has impacted on many centers of excellence in terms of HIPEC compounds, indications and perioperative management. We, therefore, aimed to depict this process at our institution and report our experience on different HIPEC drugs, means of application, indications and postoperative outcomes for patients undergoing CRS and HIPEC for gastrointestinal and gynecological primary tumors.

## 2. Materials and Methods

### 2.1. Patient Selection Criteria

From June 2005 to June 2017, a total of 399 patients underwent CRS and HIPEC for PSM of various gastrointestinal and gynecological cancers and were considered for this retrospective analysis. The patients were divided into two subgroups (group 1: 2005–2012 and group 2: 2013–2017). The main differences were the chemotherapy regimen, the application procedure (open-HIPEC and only i.p. application in group 1 and closed-HIPEC and i.v/i.p. application in group 2), indication setting and introduction of diagnostic laparoscopy pre-CRS/HIPEC.

Clinicopathological information was obtained from a prospectively collected database and electronic medical reports. The study was performed according to the guidelines of the local institutional board and the ethics committee (610/2017BO2).

Preoperative diagnostics consisted of thorough clinical examination, blood tests and a computed tomography (CT) scan. CT images were acquired with a 128-slice multidetector spiral CT. The reconstructed slice thickness was 5 mm without gaps between slices. Irresectability for CRS and HIPEC was defined as infiltration of the mesenteric axis, retroperitoneal plane or the pancreatic head. Irresectability regarding LM was dictated by metastases located in both liver lobes that were not suitable for atypical resection and that needed extended liver resection. Eligibility for CRS and HIPEC and resection of concurrent liver metastasis was assessed by a surgical oncologist, a medical oncologist, a radiologist and a radio-oncologist, all of whom attended the interdisciplinary oncologic team meeting. Adverse events were classified according to the Clavien–Dindo complication score [17].

### 2.2. Indications

Compared to group 1, patients in group 2 were only considered for CRS and HIPEC if they met the following parameters:Deemed cytoreducable on laparoscopy;Response to systemic chemotherapy;PCI cut-offs (gastric: 6–10; colorectal/appendix <16; Mesothelioma/PMP/Ovarian: no cut-off).

Patients with signet-ring histology or RAS/BRAF mutations were not excluded from CRS and HIPEC in both patient groups.

### 2.3. Cytoreductive Surgery

After laparotomy through a mid-line incision and complete adhesiolysis, the PCI was determined following the criteria described by Jaquet and Sugarbaker [16]. Abdominal regions were categorized as the small bowel, consisting of Sugarbaker’s abdominopelvic regions (SAPR) 9 to 12; the upper abdomen, consisting of SAPR 0 to 3; and the lower abdomen/pelvis consisting of SAPR 4 to 8. Tumor-involved structures were resected along with peritonectomy procedures described by Sugarbaker [18,19,20] aiming for complete cytoreduction (CC-0 and CC-1 (CC-0 indicated no visible disease; CC-1 indicated nodules smaller than 0.25 cm; CC-2 indicated nodules greater than 0.25 cm and smaller than 5 cm; CC-3 indicated nodules over 5 cm)).

### 2.4. HIPEC

After complete cytoreduction and fashioning of intestinal anastomoses, HIPEC was administered for 30 to 90 min at 42 °C, depending on the HIPEC compound, using the open- (group 1) or the closed abdomen (group 2) technique. The dosage for oxaliplatin was 300 mg/m^2^, for mitomycin C was 35 mg/m^2^ and for cisplatin was 75 mg/m^2^ body surface area. Patients receiving oxaliplatin i.p. (300 mg/m^2^; 30 min) also simultaneously received 5-FU (400 mg/m^2^) i.v. In group 1, MMC-based (90 min) or cisplatin-based (60 min) HIPEC was administered, whereas in group 2, a bi-directional chemotherapy protocol (OX (i.p.)/5-FU (i.v.) or a combination of cisplatin/doxorubicin for HIPEC was administered.

After HIPEC completion, the abdomen was washed out with 3 L of Ringer’s lactate solution and the abdomen was reopened for the removal of the perfusion catheters before the fascial closure was performed.

### 2.5. Statistics

Stata SE 13 was used for survival analysis. All analyses were stratified by group 1 and 2. Subgroups examined were PCI <17 and PCI ≥17; G2/G3; N0/N1/N2; CC0/CC1; WT and MUT. After descriptive analysis of the data (i.e., examining survival time, incidence rate and the 25, 50 and 75% survival time), log rank tests were used to examine whether the survival functions were equal across the groups. Lastly, Kaplan–Meier survival curves were used to visualize survival across the subgroups.

## 3. Results

From June 2005 to June 2017, a total of 399 patients underwent CRS and HIPEC. Group 1 contained 237 patients, while group 2 contained 162. The median age did not differ between the groups (55.3 vs. 54.2 years; *p* = 0.30). Furthermore, sex distribution was also similar (34.2% males vs. 42% males; *p* = 0.070). In both groups, peritoneal metastasis (PM) mainly originated from colorectal cancer (CRC) (28% vs. 32%). The proportion of patients with recurrent PM of ovarian origin significantly decreased when comparing group 1 and group 2, respectively (27% vs. 6%; *p* < 0.0001). The remaining tumor etiologies are listed in Table 1.

In group 1, 54% received MMC-based and 41% cisplatin-based HIPEC, whereas in group 2, 61% received a bi-directional chemotherapy protocol (OX (i.p.)/5-FU (i.v.) and 30% received a combination of cisplatin/doxorubicin for HIPEC. The median PCI for all patients was 14, with a significant decrease comparing group 1 and 2 (15.5 vs. 11; *p* = 0.002). Operative time also decreased significantly over time (541 vs. 315 min; *p* <0.0001). In group 1, in 93% of patients, a CC-0/1 resection was achieved (CC-0: 63%, CC-1: 30%). In group 2, in 100% of patients, a CC-0/1 status was attained (CC-0: 69%; CC-1: 31%) (*p* = 0.010). During the same period, 165 patients received explorative laparotomy or debulking surgery only. The number of patients treated without HIPEC declined from group 1 to group 2 (105 vs. 65 patients; *p* = 0.137).

For the achievement of a CC-0/1 score, a variety of visceral resections were necessary (Table 1). In both groups, omentectomy was carried out the most frequently (49% vs. 64%).

### 3.1. Morbidity and Mortality

The total complication rate was comparable between both groups (60% vs. 62%; *p* = 0.210). The rate of complications equal to or larger than grade III was also similar in both groups (16 % vs. 13%). Re-operation was necessary in 12% of group 1 and in 15% of group 2 and was also comparable. The most common complication in group 1 was postoperative leucopenia (34 % vs. 16%; *p* < 0.0001). In group 2, the most common complication was the occurrence of pleural effusion (10% vs. 23%; *p* = 0.091). Instances of anastomotic insufficiencies (4% vs. 6%; *p* = 0.332) and fascial rupture (3% vs. 6%; *p* = 0.588) were low and comparable between both groups. Further complications are listed in Table 1. In both groups, no in-hospital or 30-day mortality occurred. The length of hospital stay differed significantly between both groups (17 vs. 14 days; *p* < 0.0001).

### 3.2. Survival Data

The median overall survival for patients in group 1 and 2 for colon cancer, ovarian cancer gastric cancer and appendix cancer were 34 (range, 1–85) and 25 (range, 3–42) months; 45 (range, 10–142) months and not reached; 30 (range, 9–117) and 16 (range, 5–32) months) and 39 (range, 32–61) months and not reached (Table 2, Figure 1 and Figure 2), respectively.

Patients with CRC in group 2 showed a trend towards a better overall survival with low to intermediate PCI values (<17), but did not reach statistical significance (*p* = 0.183). Furthermore, in CRC for both groups, the CC score (CC-0 vs. CC-1), grading (G2 vs. G3) and lymphonodal status (N0 vs. N1 vs. N2) did not significantly impact overall survival, respectively.

## 4. Discussion

The implementation of additional flushing of the abdominal cavity with a hyperthermic chemotherapeutic solution after maximal cytoreductive surgery was the milestone in the surgical treatment of peritoneal surface malignancies. Since the first description by Spratt and co-workers in the early 1980s, a variety of experimental and clinical research has been undertaken in order to generate parameters helping surgeons to perform a strict patient selection and to provide a safe procedure to patients who are likely to benefit the most from this aggressive surgical approach [19].

Our current analysis aimed to investigate the impact of different HIPEC protocols and indication setting on long-term results, especially focusing on differing PCI and CC scores and postoperative morbidity and mortality. Strict patient selection is imperative in order to provide a low-morbidity and low-mortality procedure. This means that only patients with favorable tumor characteristics and a low tumor burden should be included in order to minimize the risk of non-therapeutic laparotomies.

A retrospective cohort, multicentric study from 23 French centers showed that the PCI and the experience of the center were statistically significantly linked to increased postoperative morbidity. Centers were classified as experienced (>7 years of practice) and as inexperienced (<7 years of practice). These data once again highlight the importance of center experience in order to provide low-morbidity CRS and HIPEC [16]. Recently, health insurance data from Germany showed that over a 9-year period, the outcome of manageable but life-threatening complications after CRS and HIPEC significantly improved, resulting in a cumulative in-hospital mortality of 3.4% [15]. Data show that at least 141 procedures have to be performed to gain sufficient expertise [20]. In the study from Huang et al., 800 patients who received CRS and HIPEC were divided into two equal groups and compared [21]. The authors stated that patients in the second group had a significantly lower PCI, a better resection status and lower morbidity and mortality rates. These results are in line with ours, which also show that patients in group 2 had a significantly lower PCI, and a significantly higher proportion of patients received a complete resection (CC-0/1). The phenomenon of a decreasing PCI over time is attributable to a better patient selection, which is expressed by our declining rate of non-therapeutic laparotomies which can partially be attributed to the implementation of pre-HIPEC laparoscopy. Hentzen et al. stated that, after implementation of routinely performed laparoscopies prior to CRS and HIPEC, a significant decrease in non-therapeutic laparotomies was experienced (21% vs. 35.4%; *p* = 0.044) [22]. Likewise, Iversen and co-workers showed that laparoscopy is a useful tool in the patient selection process, but only 17 out of 27 patients, who were deemed amenable to CRS and HIPEC by laparoscopy, were classified as resectable on laparotomy [23]. This is in line with our experience; the PCI evaluated by laparoscopy is approximately 30% too low.

In our patient cohort, the median PCI significantly decreased from 15.5 to 11 over time, which resulted in a significantly higher proportion of CC-0 resections. One of the most consistent independent predictors of morbidity is the extent of disease, measured by the PCI [24,25,26,27,28]. In our study, we also made the observation that a lower PCI was linked to fewer grade-III complications (15.6% vs. 13%). A higher PCI is linked to a more aggressive surgery, thus triggering postoperative complications. Saxena et al. analyzed 145 patients receiving CRS and HIPEC for PMP and showed that a PCI >21 and an ASA score >3 were linked to grade-IV/V morbidity [29]. Furthermore, the literature suggests that the independent contribution of HIPEC to morbidity seems to be quite low. Yang et al. compared patients with CRS only and CRS plus HIPEC in patients with PM originating from gastric cancer and found no significant difference in the occurrence of serious adverse events (11.7% vs. 14.7%; *p* = 0.839) [30]. The same results were reported by Bonnot et al., comparing CRS alone and CRS plus HIPEC for PM of gastric cancer, showing a similar major complication rate in both groups (53.7% vs. 55.3%; *p* = 0.496) [31].

For CRS and HIPEC, it is almost impossible to attribute a complication exclusively to CRS or HIPEC. In our study, we observed a HIPEC-associated morbidity which was linked to the HIPEC compound used. As in many other publications, and over time, many different chemotherapeutic agents were used with different dosages and different perfusion times, making it impossible to compare data. In our study, we found that MMC-based HIPEC is a trigger for postoperative leucopenia. In total, 50.8% of patients who received MMC-based HIPEC developed leucopenia. This is in line with other reports [32,33]. In a former study of our group, the usage of either MMC or platin compounds was not linked to an increased morbidity. In this study, CRS and HIPEC were combined with liver resection, and the exposure of the liver resection margins to intraperitoneal heated chemotherapeutic agents did not trigger bile leakage or bleeding [7,34].

In Germany, approximately 55% of HIPEC procedures were performed for CRC-PM patients in 2018 [15], and the literature provides 5-year survival rates of up to 54% in patients treated with CRS and HIPEC [35]. The impact of systemic chemotherapy on CRC-PM was evaluated by a large database analysis. Franko and co-workers reviewed 10,635 patients with metastasized CRC, but only 1.9% (*n* = 194) had CRC-PM, with a median overall survival of 16.3 months (with cytostatic agents) and 17.1 (with at least one targeted therapy) [36]. These data reveal two key facts: patients with isolated CRC-PM are highly outnumbered in clinical trials, and the effect of targeted therapy seems to be negligible. Compared to these results, the median OS of our patients was 34 and 25 months, respectively. To date, our survival rates for CRC-PM have not been reached by systemic chemotherapy. The decrease in survival in group 2 might be attributed to a high percentage of KRAS- and BRAF-mutated primary tumors compared to group 1 (12% vs. 44%).

Recently, Schneider and co-workers were the first to show that KRAS (Hazard Ratio (HR) 1.46) and BRAF (HR 3.97) mutations of the primary tumor negatively affected survival after CRS and HIPEC [37]. Another recently published study found KRAS mutations (HR 2.02) to be an independent predictor of reduced OS after CRS and HIPEC [38] and Morgan and co-workers found KRAS mutations to be associated with early recurrence after CRS and HIPEC [39]. These data suggest that mutations in genomic driver genes impact on OS after CRS and HIPEC, but underlying studies were not randomized trials and mostly included a low number of patients. Furthermore, recent studies have revealed that BRAF mutations are associated with reduced OS, but no information is provided on the proportion of peritoneal and distant recurrent disease [40,41]. For the present time, evidence is too scarce to refuse these patients CRS and HIPEC in the event of detected mutations, because the true rate of peritoneal recurrent disease in KRAS- and BRAF-mutated patients undergoing CRS and HIPEC is not known. These patients should rather be evaluated for an aggressive perioperative systemic chemotherapy in the context of CRS and HIPEC.

Recently, a German database analysis of 235 patients with PM of gastric cancer showed a median OS time of 13 months with a 5-year survival rate of 6%. The median OS differed significantly according to the PCI range (0–6: 18 months; 7–15: 12 months; 16–39: 5 months; *p* = 0.002) [42]. These data are line with ours, showing a median OS of 30 and 16 months for group 1 and 2, respectively. Likewise, the CYTO-CHIP study, comparing CRS alone and CRS and HIPEC for PM of gastric cancer, showed a prolongation of OS and recurrence-free survival (RFS) [31]. These analyses and our data outline the impact of immediate adjuvant HIPEC in the context of a very low peritoneal tumor burden on patient survival.

One main issue which was also discovered by our study was the significant decline in patients with ovarian cancer and PM. In the first group, the proportion was 27%, and in the second group, 6%. This phenomenon can be explained by the launch of the new German S3-guidelines for the therapy of ovarian cancer, which clearly disfavors the application of HIPEC. Despite the fact that reliable data on the effectiveness of additional HIPEC in ovarian cancer are available, they are still not included in the current treatment algorithm [7,43,44,45]. Of particular note are the results from Van Driel and colleagues, who compared interval CRS alone ± HIPEC and showed a significant improvement in median recurrence-free survival (10.7 vs. 14.2 months; *p* = 0.003) and median OS (33.9 vs. 45.7 months; *p* = 0.02). The fact that only 67% and 69% of patients received macroscopic complete resection [45] is also noteworthy.

Contradictory results were provided by Coleman et al. reporting a significant difference in median OS between surgery plus systemic chemotherapy versus systemic chemotherapy alone in patients with platin-sensitive recurrent ovarian cancer, in favor of systemic chemotherapy alone [46]. The data should be interpreted with caution, because 10% of the patients had additional intra- or extra-abdominal metastasis, the PCI was not provided and a highly selected patient subset (platin-free interval of 20.4 and 18.8 months) was included.

Furthermore, in a previous study of our group, patients with peritoneal and hepatic metastases of ovarian cancer were subjected to CRS, HIPEC and liver resection, and in all patients, a radical macroscopic resection was achieved with a median overall survival of 30 months. Grade-III/IV morbidity rates were 23%. These data show that even patients with ovarian cancer and PM can benefit from an aggressive surgical approach [7].

Due to the amount of clinical data favoring CRS and HIPEC for both primary and recurrent ovarian cancer, it seems increasingly incomprehensible that these patients are still refused CRS and HIPEC [47,48,49,50]. Furthermore, the ongoing criticism from gynecologists, namely, that CRS and HIPEC is a high-risk procedure, associated with high postoperative morbidity and mortality, thus delaying adjuvant systemic chemotherapy, must be invalidated, because recent data show the opposite [15,45]. The upcoming German guideline commission should, therefore, urgently redefine the clinical importance of CRS and HIPEC in ovarian cancer.

## 5. Conclusions

Strict patient selection and optimal perioperative management are crucial in order to provide low-morbidity and low-mortality CRS and HIPEC in selected patients with PM. Knowledge of patient- and treatment-related factors triggering postoperative morbidity is essential. In the near future, due to the overwhelming data favoring CRS and HIPEC for PM of ovarian cancer, selected patients should be urgently evaluated for this approach in order to further prolong OS and recurrence-free survival.

## Figures and Tables

**Figure 1 cancers-13-02471-f001:**
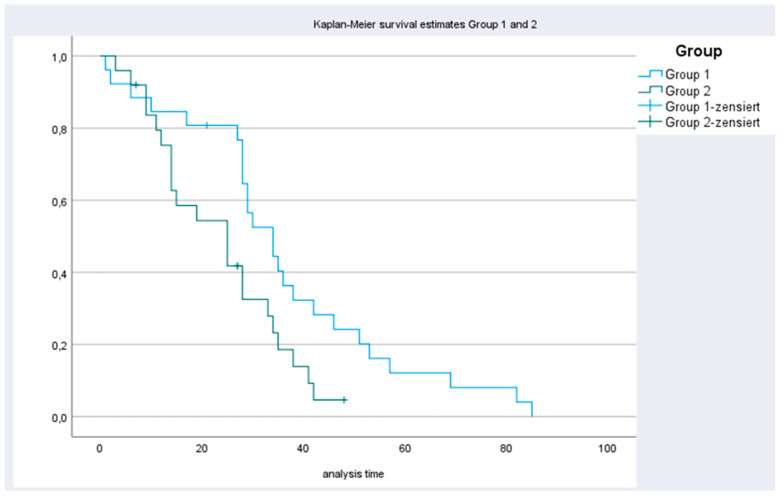
Kaplan–Meier curves for patients with CRC in group 1 and 2.

**Figure 2 cancers-13-02471-f002:**
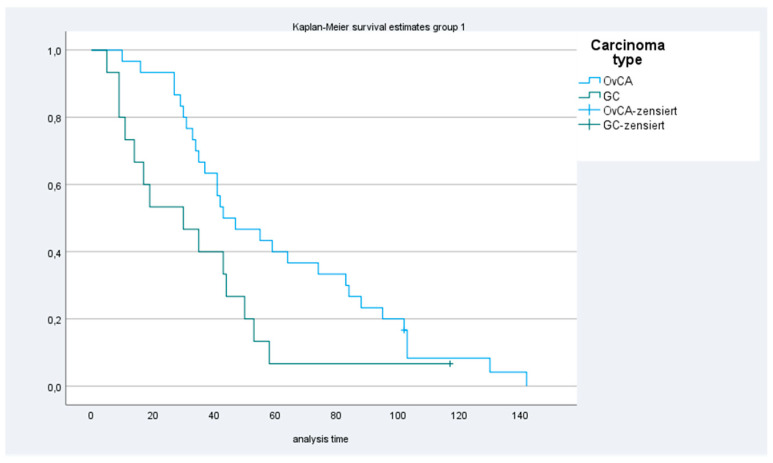
Kaplan–Meier curves for patients with ovarian cancer (OvCA) and with gastric cancer (GC).

**Table 1 cancers-13-02471-t001:** Patient- and treatment-related parameters (5-FU = 5-Fluorouracil; CC = completeness of cytoreduction; CRC = colorectal cancer; i.p. = intraperitoneal; i.v. = intravenous; min = minutes; MMC = mitomycin C; OX = oxaliplatin; PCI = peritoneal cancer index; PMP = pseudomyxoma peritonei; SSI = surgical site infections).

Parameter	Group 1 (*n* = 237)	Group 2 (*n* = 162)	*p*-Value
**Median Age (range)**	55.3 (14–75)	54.2 (19–79)	0.3
**Sex (male) %(*n*)**	34 (81)	42 (68)	0.07
**Tumor etiology %(*n*)**			
CRC	28 (67)	32 (51)	0.268
Ovarian	27 (64)	6 (10)	**<0.0001**
Gastric	12 (28)	12 (19)	0.979
Appendix	9 (21)	17 (28)	**0.011**
Mesothelioma	5 (12)	3 (5)	0.337
PMP	11 (26)	17 (28)	0.075
Others	8 (19)	13 (21)	0.648
**Median PCI (range)**	15.5 (1–39)	11 (1–39)	**0.002**
**Operative times (min)**	541 (107–1076)	315.5 (66–770)	**<0.001**
**CC-score % (*n*)**			0.010
CC-0	63 (150)	69 (112)	
CC-1	30 (71)	31 (50)	
CC-2	4 (9)	-	
CC-3	3 (7)	-	
**HIPEC technique**	open	closed	
**HIPEC compound % (*n*)**			
MMC	54 (128)	-	
Cisplatin	41 (97)	-	
MMC/Cisplatin	3 (8)	-	
OX (i.p.)/5-FU (i.v.)	-	61 (98)	
OX	-	3 (6)	
Cisplatin/Doxorubicin	-	30 (48)	
Others	2 (4)	6 (10)	
**HIPEC duration**			
**OX-based**			
**Cisplatin-based**	60 min	30 min	
**MMC-based**	90 min		
**Resections % (*n*)**			
Omentectomy	49 (116)	64 (104)	**0.003**
Appendectomy	11 (26)	20 (32)	0.339
Splenectomy	31 (73)	7 (11)	**<0.0001**
Rectum	25 (59)	12 (19)	**0.007**
Small bowel	23 (55)	13 (21)	**0.014**
Internal genitals	23 (55)	20 (32)	0.323
Right colon	17 (40)	14 (23)	0.078
Gastric	17 (40)	19 (31)	0.562
**Complication rate % (*n*)**			
Total	60 (142)	62 (101)	0.21
>Grade IIIa	16 (37)	13 (21)	0.461
**Complication type % (*n*)**			
Leucopenia	34 (81)	16 (26)	**<0.0001**
Anastomotic	4 (9)	6 (10)	0.332
insufficiency			
Pleural effusion	10 (24)	23 (37)	0.091
Pneumonia	4 (9)	3 (5)	0.704
Pulmonary embolism	6 (14)	3 (5)	0.152
Fascial rupture	3 (7)	6 (10)	0.588
SSI	10 (24)	6 (10)	0.079
Back to theatre	15 (36)	12 (20)	0.421
**30-day mortality**	0%	0%	
**Hospital stay (days (range))**	17 (3–105)	14 (6–74)	**<0.0001**

**Table 2 cancers-13-02471-t002:** Median overall survival for each tumor etiology (CRC = colorectal cancer; OS = overall survival; n.r. = not reached).

Tumor Etiology	Median OS (Months (Range))
CRC	Group 1: 34 (1–85)
	Group 2: 25 (3–42)
Ovarian	Group 1: 45 (10–142)
	Group 2: n.r.
Gastric	Group 1: 30 (9–117)
	Group 2: 16 (5–32)
Appendix	Group 1: 39 (32–61)
	Group 2: n.r.

## Data Availability

Data supporting results are harbored by an in-hospital database. Regulatory issues do not allow the provision of a link to analyzed data sets.

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
