# Peer review of "Twelve-Year Single Center Experience Shows Safe Implementation of Developed Peritoneal Surface Malignancy Treatment Protocols for Gastrointestinal and Gynecological Primary Tumors"

_cancers, 2021, doi:10.3390/cancers13102471_

Round 1

Reviewer 1 Report

How is this study truly unique compared to the studies quoted by the authors? Well written and presented but unclear how this adds to body of literature as we have seen multiple similar studies.

Author Response

Journal: Cancers

Manuscript ID: cancers-117972

Title: Development, implementation and advancement of peritoneal surface malignancy treatment protocols for gastrointestinal and gynecological primary tumors

Dear Editors.

Thank you very much for giving us the opportunity to revise our manuscript.

Below you find the answers to the comments of Reviewer #1. In the manuscript file we used the "Track Changes" function to indicate the changes made.  Additionally, the changes made a marked in red below the reviewer comments.

Reviewer #1

Comments and Suggestions for Authors

How is this study truly unique compared to the studies quoted by the authors? Well written and presented but unclear how this adds to body of literature as we have seen multiple similar studies.

Dear Reviewer. First of all, thank you for reviewing our manuscript. We will try to give you a compact answer to your question.

We think that as one of three leading peritoneal surface malignancy (PSM) treatment centers in Germany we presented our experience with nearly all issues associated with PSM treatment. It is quite obvious that, as it is a retrospective study, the manuscript never intended to be unique. The main issue of this manuscript is to guide the reader through all the hard learned lessons of PSM treatment (patient selection, pre-HIPEC laparoscopy, HIPEC protocols/technique and duration and so on..). Moreover, HIPEC protocol associated adverse events are discussed. The manuscript should provide centers of interest all major information for setting up their own PSM treatment center (apart from specialized technical aspect, i.e. technical details of an individual HIPEC-perfusion-machine). From my point of view, I would have been glad if such a detailed manuscript would have existed by the time we built up our PSM facility.

Furthermore, as also outlined by the Academic Editor, the discussion gives a quite good summary of the current literature of interest.

Reviewer 2 Report

On the one hand I do like this study, on the other hand I am a bit disappointed as there are multiple aspects preventing drawing firm conclusions. One major issue is that comparison of different protocols is tough when not performed on the same patient population. Group 1 and group 2 appear to be different in composition and disease stage as you also remark the PCI score significantly decreased in time and differs for the two groups. As it is your results are mainly useful for assessing incidence of side effects, and for showing that CRS + HIPEC is a safe procedure in the hands of an experienced high volume center. I feel that focus in title and abstract should be (even more) on feasibility and safety.

This does not mean that the authors should not attempt to go more at depth in interpreting outcome, they do not do so really much regarding the data reported in table 1 and figure 1 for instance. As such I recommend

Detailed comments:

Title of the paper is exceptionally vague, the least one would expect is an addition like ’: 12-year single center experience’. And ‘development, implementation and advancement’ is very vague, is it possible to rephrase this to something like: ’12-year single center experience shows safe implementation of developed peritoneal surface malignancy treatment protocols for gastrointestinal and gynecological primary tumors.

Page 2, line 46-50: it would be logical to mention and discuss some of the discussion on the reasons why these studies did not show the expected benefit, see for instance :

Tanis PJ, Tuynman JB, de Hingh IHJT. Results from the PROPHYLOCHIP-PRODIGE 15 trial. Lancet Oncol. 2020 Nov;21(11):e496

Ceelen W. HIPEC with oxaliplatin for colorectal peritoneal metastasis: The end of the road? Eur J Surg Oncol. 2019 Mar;45(3):400-402

Page 2, line 52: You refer to the meta analysis of Zhang et al (ref 4) which did not show differences in outcome between different protocols, but quite frankly that analysis was comparing apples and oranges, so no firm conclusions can be drawn. I suggest rephrasing to ‘suggested no evidence for differences in overall- and disease-free survival’.

Page 3, line 115/116: it is not crystal clear that you mean ‘not excluded in patients groups 1 and 2’

Page 3, line 129: you do not make clear the HIPEC duration of each protocol

Page 4, table 1: this table contains a lot of information, at the same time a lot of information is missing. I understand from the text that group 1 was treated with open HIPEC and group 2 with closed HIPEC? If yes please state so in the appropriate column along with Group 1 and Group 2 in the top row. If not then please add a separate parameter ‘HIPEC technique’ to the table. And my I request similarly to also include HIPEC duration (30, 60, 90 min?) to the table? I would think that that is relevant information. Maybe also print any variable which is statistically significant for group 1 and 2 in bold print to guide the eye.

A major improvement would be if you would present the gradual changes in tumor etiology, number of procedures, operative time etc graphically, per annum plotted against time on the horizontal axis.

Page 6, figure 1: you are not doing much in terms of interpretation with the results from Figure 1. You did not discuss in more detail why survival seems to have dropped between groups 1 and 2, is that protocol, technique or patient selection? If main focus is on feasibility, then why not move this figure to supplementary?

Page 8, line 249: ‘s’ is missing in ‘predictors’

Page 12, line 481: One author name in Ref 40 has been misspelled, you left out the ‘P’ in ‘Pelz’, the 4th author in ref 40:

Rau B, Brandl A, Piso P, Pelz J, Busch P, Demtröder C, Schüle S, Schlitt HJ, Roitman M, Tepel J, Sulkowski U, Uzunoglu F, Hünerbein M, Hörbelt R, Ströhlein M, Beckert S, Königsrainer I, Königsrainer A;

Author Response

Journal: Cancers

Manuscript ID: cancers-117972

Title: Development, implementation and advancement of peritoneal surface malignancy treatment protocols for gastrointestinal and gynecological primary tumors

Dear Editors.

Thank you very much for giving us the opportunity to revise our manuscript.

Below you find the answers to the comments of Reviewer #2. In the manuscript file we used the "Track Changes" function to indicate the changes made.  Additionally, the changes made a marked in red below the reviewer comments.

Reviewer #2

Comments and Suggestions for Authors

On the one hand I do like this study, on the other hand I am a bit disappointed as there are multiple aspects preventing drawing firm conclusions. One major issue is that comparison of different protocols is tough when not performed on the same patient population. Group 1 and group 2 appear to be different in composition and disease stage as you also remark the PCI score significantly decreased in time and differs for the two groups. As it is your results are mainly useful for assessing incidence of side effects, and for showing that CRS + HIPEC is a safe procedure in the hands of an experienced high volume center. I feel that focus in title and abstract should be (even more) on feasibility and safety. This does not mean that the authors should not attempt to go more at depth in interpreting outcome, they do not do so really much regarding the data reported in table 1 and figure 1 for instance.

Detailed comments:

1) Title of the paper is exceptionally vague, the least one would expect is an addition like ’: 12-year single center experience’. And ‘development, implementation and advancement’ is very vague, is it possible to rephrase this to something like: ’12-year single center experience shows safe implementation of developed peritoneal surface malignancy treatment protocols for gastrointestinal and gynecological primary tumors.

You are right when saying that the title is rather vague, but this was chosen intentionally. For me as a reader a somehow vague title makes me even more attracted to read the manuscript. Nevertheless, this only my personal view and the title you suggested instead is of course more straight forward. The authors do now have a major issue with that so we are going to adapt the title.

New title of the manuscript: “12-year single center experience shows safe implementation of developed peritoneal surface malignancy treatment protocols for gastrointestinal and gynecological primary tumors.”

2) Page 2, line 46-50: it would be logical to mention and discuss some of the discussion on the reasons why these studies did not show the expected benefit, see for instance :

Tanis PJ, Tuynman JB, de Hingh IHJT. Results from the PROPHYLOCHIP-PRODIGE 15 trial. Lancet Oncol. 2020 Nov;21(11):e496

Ceelen W. HIPEC with oxaliplatin for colorectal peritoneal metastasis: The end of the road? Eur J Surg Oncol. 2019 Mar;45(3):400-402

Thanks for the hint. These two publications truly give a good insight into why these studies maybe did not deliver the results expected. We therefore included them in the introduction (see below)

“Above all the PRODIGE-7 trial was criticized due to a variety of methodological weaknesses including an expected overoptimistic survival in the experimental arm combined with an underestimation of the effect of cytoreductive surgery alone [4], Furthermore, the short drug exposure time of OX, the possible adverse effects of the carrier solution (Dextrose 5%) and possible adverse effects of hyperthermia itself were brought up explaining the negative results of the study. The same is true for the PRODIGE-15 study, which was manly criticized due to a very heterogenous study population (non-metastatic and metastatic setting) [5].”

3) Page 2, line 52: You refer to the meta analysis of Zhang et al (ref 4) which did not show differences in outcome between different protocols, but quite frankly that analysis was comparing apples and oranges, so no firm conclusions can be drawn. I suggest rephrasing to ‘suggested no evidence for differences in overall- and disease-free survival’.

Thank you for the comment. You are right, this section should be rephrased. We changed the paragraph according to your recommendation.

“Till today no randomized controlled trial comparing the efficacy of different HIPEC-compounds exists but a recent meta-analysis, including 11 studies and 2091 patients, comparing MMC- and OX-based HIPEC, suggested no evidence for differences in overall- and disease-free survival but a statistically significantly increased rate of major complications to the disadvantage of OX [4].”

4) Page 3, line 115/116: it is not crystal clear that you mean ‘not excluded in patients groups 1 and 2’

Sorry, for that. That paragraph is truly not 100% clear. What we wanted to say is that despite the presence of a RAS/BRAF-mutation or a signet-ring histology, the patients were offered CRS and HIPEC as long as other clinical parameters were fulfilled.

Revised paragraph: “Patients with signet-ring-histology or RAS/BRAF-mutations were not excluded from CRS and HIPEC in both patients groups”

5) Page 3, line 129: you do not make clear the HIPEC duration of each protocol

You are right. This information is lacking.

Revised paragraph:

“HIPEC

After complete cytoreduction and fashioning of intestinal anastomoses, HIPEC was administered for 30 to 90 minutes at 42°C depending on the HIPEC compound using the open (group 1) or the closed-abdomen (group 2) technique. The dosage for oxaliplatin was 300mg/m², for mitomycin C 35mg/m² and for cisplatin 75mg/m² body surface area. Patients receiving oxaliplatin i.p (300mg/m²; 30 minutes) also received simultaneously 5-FU (400mg/m²) i.v. In group 1 MMC-based (90 minutes) or cisplatin-based (60 minutes) HIPEC, whereas in group 2 a bi-directional chemotherapy protocol (OX (i.p.)/5-FU (i.v.) or a combination of cisplatin/doxorubicin for HIPEC was administered. “

6a)

Page 4, table 1: this table contains a lot of information, at the same time a lot of information is missing. I understand from the text that group 1 was treated with open HIPEC and group 2 with closed HIPEC? If yes please state so in the appropriate column along with Group 1 and Group 2 in the top row. If not then please add a separate parameter ‘HIPEC technique’ to the table. And my I request similarly to also include HIPEC duration (30, 60, 90 min?) to the table? I would think that that is relevant information. Maybe also print any variable which is statistically significant for group 1 and 2 in bold print to guide the eye.

Yes, you are right that the table contains a lot of information but we think it is structured quite easy. The HIPEC-technique column was added (group1 = open; group 2= closed). Furthermore, the HIPEC duration was added for each HIPEC compound. Statistically significant variables were printed in bold.

6b)

A major improvement would be if you would present the gradual changes in tumor etiology, number of procedures, operative time etc graphically, per annum plotted against time on the horizontal axis.

You are right. Therefore, we included in the original manuscript the “Graphical Abstract”

See below:

7) Page 6, figure 1: you are not doing much in terms of interpretation with the results from Figure 1. You did not discuss in more detail why survival seems to have dropped between groups 1 and 2, is that protocol, technique or patient selection? If main focus is on feasibility, then why not move this figure to supplementary?

Thank you for the comment. This in line with comment and suggestion from the Academic Editor. We added Kaplan-Meier curves for patients with ovarian and gastric cancer. Furthermore, the section “survival data” in the results section was reorganized (see below). According to the suggestion of the Academic editor the results of biomarkers were eliminated.

Survival data

Median overall survival for patients in group 1 and 2 for colon cancer, ovarian cancer gastric cancer and appendix cancer were 34 (range, 1-85) and 25 (range, 3-42) months; 45 (range, 10-142) months and not reached; 30 (range, 9-117) and 16 (range, 5-32) months) and 39 (range, 32-61) months and not reached (Table 2, Figure 1, 2 and 3), respectively.

Tumor etiology

Median OS (months (range))

     CRC

Group 1: 34 (1-85)

Group 2: 25 (3-42)

     Ovarian

Group 1: 45 (10-142)

Group 2: n.r.

     Gastric

Group 1: 30 (9-117)

Group 2: 16 (5-32)

Appendix

Group 1: 39 (32-61)

Group 2: n.r.

Table 2: Median overall survival for each tumor etiology (CRC= colorectal cancer; OS= overall survival; n.r.= not reached)

Figure 3: Kaplan-Meier curves for patients with ovarian cancer (OvCA) and with gastric cancer (GC)

Kaplan-Meier curves for patients with CRC in group 2 (Figure 2) showed a trend towards a better overall survival for patients with low to intermediate PCI-values (<17) but without reached statistical significance (p=0.183). Furthermore, in CRC for both group 1 and 2 the CC-score (CC-0 vs. CC-1), grading (G2 vs. G3) and lymphonodal status (N0 vs. N1 vs. N2) did not significantly impact overall survival, respectively.

7a) You did not discuss in more detail why survival seems to have dropped between groups 1 and 2, is that protocol, technique or patient selection?

You are right. We do not think that it can be attributed to the protocol or the technique. Furthermore, patient variables also do not indicate major differences between groups. Only our subgroup analysis regarding mutational status in CRC indicated a higher amount of driver mutations in group 2, so that maybe this circumstance might be an explanation for the reduced OS in group 2.

As far as gastric cancer is concerned, the indications were very careful at the beginning with the occurrence of two long-term survivor. When excluding them from OS-analysis the median OS is 24,5 months, with is very good in line with the results from Rau et al.

In Germany approximately 55% of HIPEC-procedures were performed for CRC-PM in 2018 [13] and literature provides 5-year survival rates of up to 54% in patients treated with CRS and HIPEC [33]. The impact of systemic chemotherapy on CRC-PM was evaluated by a large database analysis. Franko and co-workers reviewed 10.635 patients with metastasized CRC, but only 1.9% (n=194) had CRC-PM with a median overall survival of 16.3 months (with cytostatic agents) and 17.1 (with at least one targeted therapy) [34]. These data reveal two key facts: patients with isolated CRC-PM are highly outnumbered in clinical trials and the effect of targeted therapy seems to be negligible. Compared to these results the median OS of our patients was 34 months and 25 months, respectively. Our survival rates for CRC-PM are not reached by systemic chemotherapy, so far. The drop of survival in group 2 might be attributed to a high percentage of KRAS- and BRAF-mutated primary tumors compared to group 1.

8) Page 8, line 249: ‘s’ is missing in ‘predictors’

Sorry for that.

“In our patient cohort median PCI significantly decreased from 15.5 to 11 over time which also resulted in significantly a higher proportion of CC-0 resections. One of the most consistent independent predictors of morbidity is the extent of disease, measured by the PCI [22, 23, 24, 25, 26].”

9) Page 12, line 481: One author name in Ref 40 has been misspelled, you left out the ‘P’ in ‘Pelz’, the 4th author in ref 40:

Rau B, Brandl A, Piso P, Pelz J, Busch P, Demtröder C, Schüle S, Schlitt HJ, Roitman M, Tepel J, Sulkowski U, Uzunoglu F, Hünerbein M, Hörbelt R, Ströhlein M, Beckert S, Königsrainer I, Königsrainer A;

Sorry for that.

Revised reference No. 40:

“40. Rau B, Brandl A, Piso P, Pelz J, Busch P, Demtröder C, Schüle S, Schlitt HJ, Roitman M, Tepel J, Sulkowski U, Uzunoglu F, Hünerbein M, Hörbelt R, Ströhlein M, Beckert S, Königsrainer I, Königsrainer A. Peritoneum Surface Oncology Group and members of the StuDoQ|Peritoneum Registry of the German Society for General and Visceral Surgery (DGAV). Peritoneal metastasis in gastric cancer: results from the German database. Gastric Cancer. 2020;23(1):11-22.”

Round 2

Reviewer 2 Report

I thank the authors for taking note of and processing my suggestions and recommendations. I now recommend acceptance

Author Response

Thank you very much for reviewing our manuscript.